# Iron Hydroxide/Oxide-Reduced Graphene Oxide Nanocomposite for Dual-Modality Photodynamic and Photothermal Therapy In Vitro and In Vivo

**DOI:** 10.3390/nano11081947

**Published:** 2021-07-28

**Authors:** Wei-Jane Chiu, Yi-Chun Chen, Chih-Ching Huang, Lingyan Yang, Jiantao Yu, Shih-Wei Huang, Chia-Hua Lin

**Affiliations:** 1Department of Bioscience and Biotechnology, National Taiwan Ocean University, Keelung 20224, Taiwan; abbcd60228@hotmail.com (W.-J.C.); huanging@mail.ntou.edu.tw (C.-C.H.); 2Department of Biotechnology, National Formosa University, Yunlin 63208, Taiwan; a12345314@yahoo.com.tw; 3Center of Excellence for the Oceans, National Taiwan Ocean University, Keelung 20224, Taiwan; 4School of Pharmacy, College of Pharmacy, Kaohsiung Medical University, Kaohsiung 80708, Taiwan; 5Key Laboratory of Nano-Bio Interface, Suzhou Key Laboratory for Nanotheranostics, Division of Nanobiomedicine, Suzhou Institute of Nano-Tech and Nano-Bionics, Chinese Academy of Sciences, Suzhou 215123, China; lyyang2013@sinano.ac.cn (L.Y.); jtyu2013@sinano.ac.cn (J.Y.); 6Department of Electronics, Cheng Shiu University, Kaohsiung 83347, Taiwan; envhero@gcloud.csu.edu.tw; 7Center for Environmental Toxin and Emerging-Contaminant Research, Cheng Shiu University, Kaohsiung 83347, Taiwan; 8Super Micro Research and Technology Center, Cheng Shiu University, Kaohsiung 83347, Taiwan

**Keywords:** photodynamic therapy, photothermal therapy, reduced graphene oxide, near-infrared laser, iron hydroxide/oxide

## Abstract

Minimal invasive phototherapy utilising near-infrared (NIR) laser to generate local reactive oxygen species (ROS) and heat has few associated side effects and is a precise treatment in cancer therapy. However, high-efficiency and safe phototherapeutic tumour agents still need developing. The application of iron hydroxide/oxide immobilised on reduced graphene oxide (FeOxH–rGO) nanocomposites as a therapeutic agent in integration photodynamic cancer therapy (PDT) and photothermal cancer therapy (PTT) was discussed. Under 808 nm NIR irradiation, FeOxH–rGO offers a high ROS generation and light-to-heat conversion efficiency because of its strong NIR absorption. These phototherapeutic effects lead to irreversible damage in FeOxH–rGO-treated T47D cells. Using a tumour-bearing mouse model, NIR ablated the breast tumour effectively in the presence of FeOxH–rGO. The tumour treatment response was evaluated to be 100%. We integrated PDT and PTT into a single nanodevice to facilitate effective cancer therapy. Our FeOxH–rGO, which integrates the merits of FeOxH and rGO, displays an outstanding tumoricidal capacity, suggesting the utilization of this nanocomposites in future medical applications.

## 1. Introduction

Phototherapy of solid tumours is an attractive method for non-invasive treatment [1,2,3]. Phototherapy typically involves two unique properties of photosensitisers: generating toxic ROS (reactive oxygen species), PDT (photodynamic therapy) or heat (photothermal therapy, PTT) capable of killing cancer cells via photoablation [4]. As photosensitisers are typically harmless without light, selective illumination allows precise tumour treatment, reducing side effects to healthy tissues [4]. However, numerous current PDT photosensitisers have been excited by visible light or ultraviolet (UV), limiting penetration depth and cancer treatment efficiency [3]. Additionally, PDT has been proven to damage vascular endothelial cells in the therapeutic process [5]. PTT is a hopeful technique for cancer treatment using non-poisonous light-responsive materials, which is favourable over traditional treatment, such as surgical operation and chemotherapy due to the spatio-temporal control strategy as well as reduced complications [2,6,7,8].

Ideal PTT or PDT agents should exhibit a strong absorbance at an NIR (near-infrared) of 780–1100 nm (in which less light is absorbed by tissues), allowing the efficient conversion of optical energy into thermal energy [9,10]. In recent years, nanomaterials-based photothermal agents have been extensively researched, such as carbon-based nanoparticles, gold nanoparticles and inorganic/organic nanocomposites with a strong absorbance in the NIR window. These photothermal agents effectively convert NIR laser into heat to wipe out tumour cells under NIR light illumination [8,11,12,13,14,15,16,17,18]. Additionally, some researchers have employed multi-nanomaterials to achieve a synergetic PDT/PTT effect into a single platform. This combination results in enhanced cancer-cell killing and decreases the side effects [13,17,18]. Nevertheless, most of these therapeutic strategies involve multiple steps and are complex. Moreover, some systems need more than one light source supplying sufficient energy to stimulate the PDT and PTT agents [19,20,21,22]. Multi-laser treatments are costly and time-consuming, limiting practicability in the clinic.

Graphene oxide (GO)-based materials have been extensively researched for PTT agents; these materials exhibited an excellent photothermal effect because of their rapid light-to-heat conversion compared to other allotropes of carbon [23,24]. This GO-based phototherapeutic platform also utilizes GO or reduces GO (rGO) as a carrier to transport functional composites to enhance therapy efficacy [23,24]. Previously, we reported that iron oxide–hydroxide and iron hydroxide were immobilised on rGO to produce FeOxH–rGO. This FeOxH–rGO nanocomposite exhibited high catalytic activity, decomposing hydrogen peroxide (H_2_O_2_) to reactive hydroxyl and hydroperoxyl radical via Fenton-based reactions [25]. Thus, we believe that FeOxH–rGO nanocomposites could be an ideal combination PTT and PDT therapeutic agent for cancer treatment.

Here, we investigated the therapeutic effects of FeOxH–rGO nanocomposites in vitro and in vivo. FeOxH–rGO nanocomposites exhibited high photothermal conversion and ROS production efficiency. Furthermore, we demonstrated the therapeutic effects of FeOxH–rGO nanocomposites for solid tumours in vivo in mice bearing tumours; an apparently complete resorption of the tumour was achieved with insignificant toxicity. Altogether, FeOxH–rGO exhibited high biocompatibility and excellent anticancer activity, indicating great potential for application in the field of tumour phototherapeutic treatment.

## 2. Materials and Methods

### 2.1. Chemicals

Hydrochloric acid, phosphoric acid, boric acid, tris(hydroxymethyl)aminomethane (Tris) and metal salts sulfuric acid were obtained from J.T.Baker (Phillipsburg, NJ, USA). Graphite (7–11 μm), potassium permanganate and sodium sulphide were purchased from Alfa Aesar (Tewksbury, MA, USA). H_2_O_2_ was obtained from SHOWA (Tokyo, Japan). Phosphate buffered saline, RPMI 1640 medium and foetal bovine serum (FBS) were obtained from Gibco (Waltham, MA, USA). Thiazolyl blue tetrazolium bromide (MTT) and 2′,7′-dichlorofluorescin diacetate (DCFH-DA) were obtained from Sigma-Aldrich (Saint Louis, MO, USA). High-quality ultrapure water (Billerica, MA, USA) was utilized in this research.

### 2.2. Preparation and Characterisation of FeOxH–rGO

GO was synthesised according to Hummers’ method with modification [26]. Graphite flakes and potassium permanganate were added to a mixture of phosphoric acid and sulfuric acid. The mixture was heated, stirred, cooled and then poured into H_2_O_2_ solution. The aqueous mixture was then centrifuged and the supernatant was decanted. The pellet was washed with deionized (DI) water, sonicated and centrifuged. The GO solution was collected and determined to be ~3.6 g L^−1^. The rGO was synthesised from irradiation GO under UV lamp. For preparation of FeOxH–rGO, FeCl_2_ was mixed with GO in a Tris–borate solution. The resulting FeO(OH) and Fe(OH)_2_ were immobilised on rGO. Transmission electron microscopy (TEM) was performed using an HT-7700 system (Tokyo, Japan). The Raman spectra and optical properties of FeOxH–rGO were recorded by Shimadzu UV–VIS spectrometer (Kyoto, Japan) or LabRAM HR spectrometer (Kyoto, Japan).

### 2.3. Cell Cultures

Human ductal breast epithelial tumour (T47D) cells and mice breast tumour (4T1) cells were maintained in RPMI 1640 (10% FBS) in 5% CO_2_ at 37 °C. RPMI 1640 was replaced every 3 days, and the cells were passaged twice a week by trypsinization.

### 2.4. Temperature Monitoring and ROS Generation Experiments

FeOxH–rGO suspensions in RPMI 1640 medium were irradiated with NIR laser (808 nm). The temperature elevations of FeOxH–rGO were determined by thermocouple. FeOxH–rGO or rGO solutions were added to the T47D cells. T47D cells were irradiated with NIR laser (1.82 W/cm^2^) for 5 min, and then the RPMI 1640 medium was changed with DCFH-DA solution. T47D cells were incubated for 30 min in the dark. ROS generation were measured by fluorescence spectrophotometer (λ_ex_ = 488 nm and λ_em_ = 535 nm). H_2_O_2_ (100 μM) treatment was positive control.

### 2.5. Combined PTT and PDT Therapy In Vitro and In Vivo

After FeOxH–rGO incubation, the T47D cells (8 × 10^3^) were irradiated with NIR light (1.82 W/cm^2^) for 5 min. Cell viability was determined 12 h after NIR irradiation by MTT assay. The cell viability was determined by a spectrophotometer (Twinsburg, OH, USA) at 490 nm.

To develop the tumour model, 4T1 cells (1 × 10^6^) were subcutaneously injected into the back of Balb/c female mice. The mice were divided into five groups (n = 5) after the tumour volume reached ~100 mm^3^. Each group of mice were intratumorally injected with PBS or FeOxH–rGO. Then, tumours were irradiated with or without NIR light (1.3 W/cm^2^) for 5 min. The length and width of the tumours were monitored by a digital caliper. The tumour volume was determined according to the following formula: length × width^2^/2. Relative tumour volumes were calculated as V/V0 (V0 was the initiated tumour volume). For IR thermal imaging, mice injected with FeOxH–rGO were anaesthetised and imaged under an Infrared Cameras. Inc. infrared (IR) thermal camera (Beaumont, TX, USA) with or without illumination under NIR laser (808 nm, 1.3 W/cm^2^) for 5 min.

### 2.6. Histology Examinations

FeOxH–rGO-treated and untreated Balb/c female mice bearing 4T1 tumours were sacrificed 6 h after NIR irradiation. Tumour tissues of Balb/c female mice were collected, fixed (4% formalin) and conducted (paraffin embedded sections). The samples were then stained with haematoxylin and eosin (H&E). Finally, the tumour tissues were examined under a digital microscope (Munich, Germany).

### 2.7. Statistical Analysis

All data were presented as mean ± standard deviation. Statistical Analysis was performed using one-way analysis of variance (ANOVA), followed by Dunnett test. *p* value < 0.05 was considered statistically significant.

## 3. Results and Discussion

### 3.1. Synthesis and Characterisation of FeOxH–rGO Nanocomposites

We described the synthesis and characterisation method of FeOxH–rGO [25]. Nano-FeOxH were distributed on the surface of rGO (Appendix A) (size ~300 nm; prepared from irradiation of GO with UV light for 5 h), forming FeOxH–rGO (Figure 1A). The UV–VIS absorption spectra of GO showed a broad absorption band in the UV region (Figure 1B). The absorption band at 230 nm was attributed to the π → π* transition of the C=C bond in the sp^2^ hybrid region. The absorption band at 300 nm was attributed to the n → π* electronic transition of the peroxide and/or epoxide functional groups. A significantly stronger absorption of rGO indicated that some oxygen-containing carbons were reduced to C=C (Figure 1B) [6,27]. Furthermore, the absorbance of FeOxH–rGO was enhanced in the NIR window (Figure 1B), potentially as a result of FeOxH deepening the colour (Figure 1C) and increasing light absorption, suggesting FeOxH–rGO might possess high phototherapeutic potential in cancer treatment [18]. The Raman spectra of FeOxH–rGO showed the disorder band associated with graphene edges (D band) at 1350 cm^−1^ and the specific band associated with the in-phase vibration of the graphene lattice (G band) at 1585 cm^−1^ (Figure 1D) [28]. Previously, we found that FeOxH–rGO exhibited great catalytic activity, suggesting the FeOxH–rGO could show bimodality for the dual purposes of PTT and PDT in vitro and in vivo.

### 3.2. Photothermal and Photodynamic Properties of FeOxH–rGO Nanocomposites

A photothermal conversion via various concentrations of Fe^2+^ ions to form iron(III) were investigated. We demonstrated that 100–400 μM of Fe^2+^ ions react with rGO (300 μg/mL) in Appendix A. The FeOxH–rGO nanocomposites showed superior photothermal heating to rGO at all Fe^2+^ ion concentrations (Appendix A). At a concentration of 100 μM, Fe^2+^ ions immobilised on rGO; NIR irradiation for 5 min caused the highest temperature increase (~15 °C) in the FeOxH–rGO nanocomposite solution (Appendix A). After NIR irradiation (5 min), at a concentration of 100 μM, Fe^2+^ ions immobilised on rGO and showed a great physiological stability (Appendix A). A high concentration of Fe^2+^ ions (300 and 400 μM) showed slight aggregation after NIR irradiation (Appendix A). This aggregation may have been due to the increased concentration of Fe^2+^ ions forming an excess of FeOxH and being fully immobilised on rGO, which may reduce the solubility of rGO (Appendix A). According to the above data, we employed the FeOxH–rGO nanocomposites which were prepared from 100 μM Fe^2+^ and reacted with rGO (300 μg/mL) to further experiments. To examine the photothermal effect, different concentrations of FeOxH–rGO (from 50 to 300 μg/mL) were irradiated under NIR laser (808 nm, 1.82 W/cm^2^) for 5 min. The temperatures of these FeOxH–rGO solutions increased with illumination time and the concentration of FeOxH–rGO (Figure 2A,B). After irradiation for 5 min, the temperature of the medium solution was increased by 44.0 °C, FeOxH–rGO nanocomposites at a concentration of 300 μg/mL (Figure 2B). In contrast, the temperature of the medium increased by only 0.7 °C (Figure 2B). The irradiated power-dependent photothermal heating effect (from 1.82 W/cm^2^) was also observed, which indicated the temperature increased with NIR laser (Figure 2C). With the various NIR lasers, the heating curves also showed a laser power-dependent photothermal effect for FeOxH–rGO nanocomposites (300 μg/mL) with the highest temperature increment up to 60 °C under an NIR laser power density of 2.74 W/cm^2^ irradiation for 5 min (Figure 2C). The human tumour tissues can be heated to over 50 °C within 5 min under NIR laser irradiation, efficiently killing the cancer cells after injection of FeOxH–rGO nanocomposites [29]. Additionally, the photothermal conversion efficiency (η) of FeOxH–rGO was determined using the method in SI1. The aqueous dispersion of FeOxH–rGO exhibited the highest temperature of 38.2 °C when the surrounding temperature was at 25 °C (Figure 2D). The photothermal conversion efficiency of FeOxH–rGO was estimated to be 64.4% (SI1). The η of FeOxH–rGO was much higher than other phototherapeutic agents [6,30].

We used DCFH-DA to measure the ROS accumulation in T47D cells after exposure to FeOxH–rGO (50–300 μg/mL) under 1.82 W/cm2 NIR irradiation. Nonfluorescent-DCFH-DA can be oxidised to fluorescent dichlorodihydrofluorescein (DCF) by intracellular ROS [31]. In FeOxH–rGO-treated cells, the ROS accumulation was increased by ~2–3 fold (Figure 3). Additionally, FeOxH–rGO caused severe oxidative stress to T47D cells (~2–16 fold) under NIR irradiation (1.82 W/cm^2^) (Figure 3). The modification of rGO using FeOxH increased the potential of FeOxH–rGO for PDT/PTT in cancer treatment (Figure 2 and Figure 3).

### 3.3. In Vitro Photothermal and Photodynamic Therapy of FeOxH–rGO Nanocomposites

To confirm the phototherapy efficacy of the FeOxH–rGO nanocomposites in vitro, the relative T47D cell viability was estimated before and after NIR irradiation (1.82 W/cm^2^) using the MTT assay (Figure 4). The T47D cells with only medium were the control. The FeOxH–rGO nanocomposites without 808 nm NIR laser irradiation had no cytotoxicity on T47D cell (50–300 μg/mL); thus, ensuring a large application potential in cancer phototherapy (Figure 4A). However, the relative viability of the T47D cells treated with a concentration at 300 μg/mL of FeOxH–rGO in the presence of NIR laser for 5 min decreased remarkably—72% compared to control—and showed insignificant cytotoxicity before NIR irradiation. This result indicated FeOxH–rGO nanocomposites are a safe phototherapy agent and could increase the mortality of cells upon laser irradiation (Figure 4A). To gain further insight into the mechanism responsible for the photodynamic therapy of T47D cells, MTT assay also analysed T-47D cells which were pre-cultured with FeOxH–rGO nanocomposites and pre-treated with ascorbic acid for 30 min before irradiation (Figure 4B). Cells incubated with medium only were used as the control and rGO as a comparison. The relative cell viability of T47D cells pre-cultured with FeOxH–rGO nanocomposites revealed a significant difference in the ascorbic acid concentration, which showed an increase in the relative cell viability for T47D cells upon NIR irradiation (Figure 4B). This observation concluded that ROS play an influential role in the phototherapy of cells by FeOxH–rGO nanocomposites. This nanocomposite might enable attachment onto the cell membrane and then internalize through endocytosis by tumour cells [6,23,32]. FeOxH–rGO nanocomposites are safe and effective phototherapy agents in vitro.

### 3.4. In Vivo Photothermal and Photodynamic Therapy

We performed in vivo tests to evaluate the efficacies of FeOxH–rGO mediating PDT/ PTT effects on the destruction of 4T1 cells tumours (~100 mm^3^). As shown in Figure 5A, each group of mice (n = 5) was intratumorally injected with 20 μL PBS or FeOxH–rGO (2.4 mg/mL). Then, those tumours were irradiated with 808 nm NIR laser (1.3 W/cm^2^) for 5 min. Mice with an intratumoral injection of PBS or FeOxH–rGO nanocomposites without NIR illumination were also used as the control. Temperature changes of mice under NIR irradiation were monitored by an IR thermal camera (Figure 5B(a)). The temperatures of the control group were still below or around 40 °C (Figure 5B(a)). In contrast, more significant temperature increases were noticed for tumours injected with FeOxH–rGO nanocomposites, reaching temperatures of 56.1 °C (Figure 5B(a)). After various treatments, we investigated the phototherapeutic effect of cancer in Balb/c mice bearing 4T1 tumours in the next 17 days. The results showed that the tumours treated with PBS and FeOxH–rGO with laser irradiation grew rapidly, suggesting the 4T1 tumour growth was not affected by NIR laser or FeOxH–rGO alone (Figure 5B(b)). For mice treated with FeOxH–rGO plus NIR laser irradiation, no tumour recurrence was observed in the tumour site, and the original tumour sites were restored and hair grew rapidly about 2 weeks later (Figure 5B). The volume of the 4T1 tumour in mice was not affected when treated with PBS, PBS with laser or FeOxH–rGO alone (Figure 5B(b) and Figure 6A,B). However, FeOxH–rGO injection coupled with NIR illumination significantly removed tumour and no tumour recurrence was observed. Photographs more clearly demonstrate this effect in Figure 6C.

The length and width of the tumours were also monitored by a digital caliper every 2 days over the next 17 days (Figure 7A). The body weights of mice were not significantly varied, indicating no side effects of our phototherapy (Figure 7B). In addition, the tumour tissue was examined histologically to evaluate the PDT and PTT obtained by FeOxH–rGO injection and NIR laser illumination. As shown in Figure 5A, tumour cells treated with FeOxH–rGO injection and NIR laser illumination were heavily damaged, demonstrating the superiority of this PDT/PTT combination therapy.

## 4. Conclusions

We have developed an amorphous FeOxH–rGO for dual-modality PDT and PTT via the reaction of GO partially reduced by UV irradiation with an Fe^2+^ solution. We demonstrated that FeOxH–rGO generate ROS via Fenton based-reactions and high light-to-heat conversion under NIR irradiation. Both in vitro and in vivo experiments demonstrate feasibility for use as a phototherapeutic agent in cancer therapy. Notably, this study showed that FeOxH–rGO actively removed tumours with no recurrence or acute side effects. This study provides a facile method to develop efficacious dual-modality GO-based nano-platform for cancer thermo-therapeutics.

## Figures and Tables

**Figure 1 nanomaterials-11-01947-f001:**
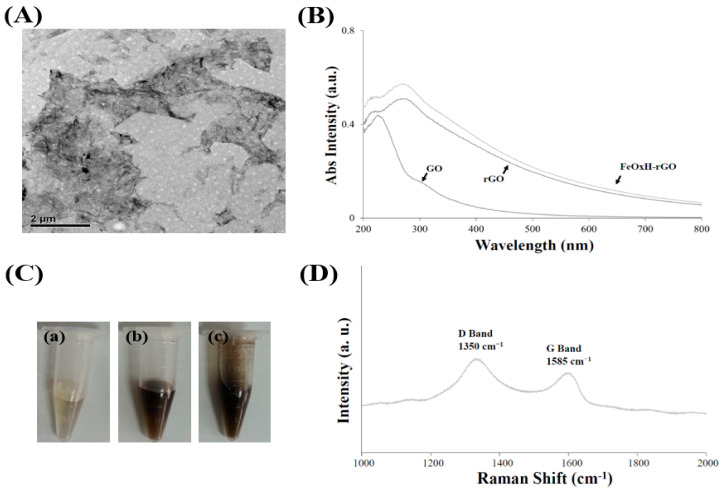
Physicochemical properties of FeOxH–rGO nanocomposites. (**A**) TEM images of FeOxH–rGO. (**B**) The UV–VIS absorption spectra of GO, rGO and FeOxH–rGO. (**C**) Photographs of the (**a**) GO, (**b**) rGO and (**c**) FeOxH–rGO solutions. (**D**) Raman spectra of FeOxH–rGO nanocomposites. Absorbance (Abs) is plotted in arbitrary units (a.u.).

**Figure 2 nanomaterials-11-01947-f002:**
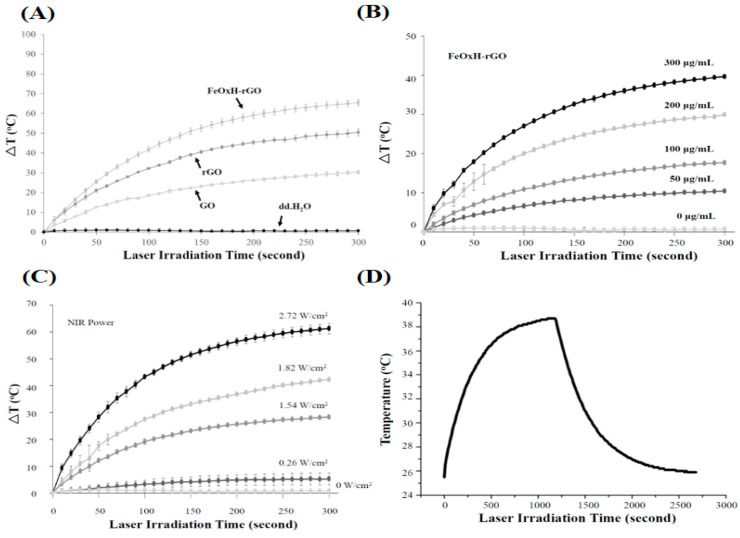
Photothermal properties of FeOxH–rGO nanocomposites. (**A**) Photothermal heating of dd. H_2_O, GO, rGO and FeOxH–rGO under NIR illumination (880 nm, 2.72 W/cm^2^). (**B**) Photothermal heating of different concentrations of FeOxH–rGO (0–300 μg/mL) under NIR illumination. (**C**) Photothermal heating of FeOxH–rGO (300 μg/mL) under different intensities of NIR illumination (880 nm, 0–2.72 W/cm^2^). (**D**) Photothermal conversion efficiency of FeOxH–rGO under NIR illumination.

**Figure 3 nanomaterials-11-01947-f003:**
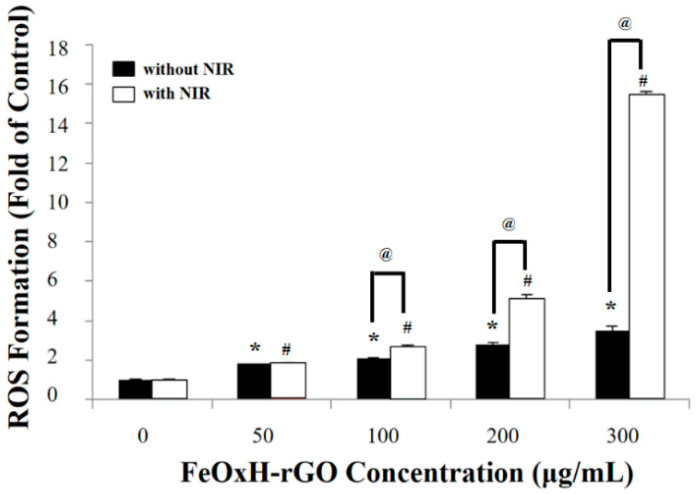
Photodynamic properties of FeOxH–rGO nanocomposites. ROS formation of FeOxH–rGO (0–300 μg/mL) with or without NIR illumination (880 nm, 2.72 W/cm^2^). * *p* < 0.05 and # *p* < 0.05 indicate statistically significant differences from the control (0 μg/mL). @ *p* < 0.05 indicates statistically significant differences from FeOxH–rGO-treated T47D cells without NIR illumination.

**Figure 4 nanomaterials-11-01947-f004:**
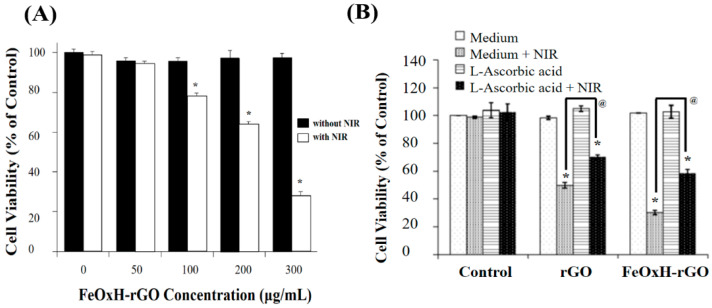
In vitro phototherapeutic effect of FeOxH–rGO nanocomposites. (**A**) Cell viability of T47D cells after treatment with various concentrations of FeOxH–rGO with or without laser irra-diation for 5 min. (**B**) Cell viability of T47D cells after incubation with medium, rGO, FeOxH–rGO and L-ascorbic acid, followed by NIR irradiation. The power density of NIR used in this set of ex-periments is 1.82 W/cm^2^ and irradiation for 5 min. * *p* < 0.05 indicates statistically significant dif-ferences from the control. @ *p* < 0.05 indicates statistically significant differences from rGO-treated or FeOxH–rGO-treated T47D cells with NIR illumination.

**Figure 5 nanomaterials-11-01947-f005:**
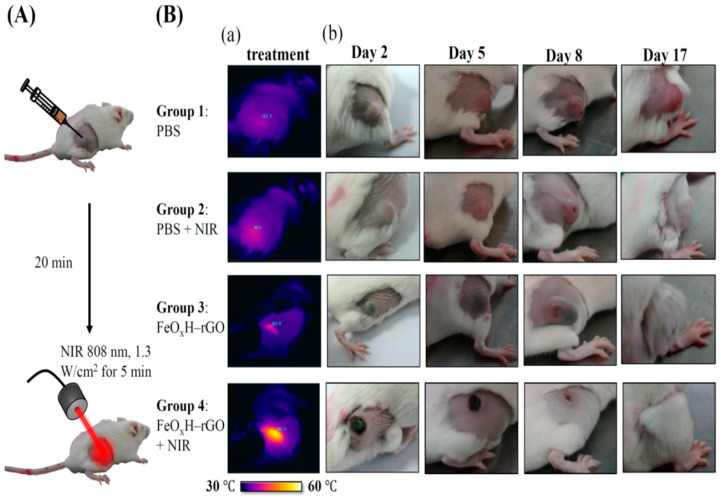
In vivo phototherapeutic effect of FeOxH–rGO nanocomposites. (**A**) Scheme of combination therapy based on intratumorally injected FeOxH–rGO. (**B**) (**a**) IR thermal images of 4T1 tumour-bearing mice recorded by an IR camera, with or without NIR laser irradiation of 5 min, and (**b**) representative photos of mice bearing 4T1 tumours 17 days after treatment. Combined treatment of FeOxH–rGO + NIR laser yielded higher synergistic therapy effect and no tumour recurrence was noted over a course of 17 days. Group 1: PBS; Group 2: PBS + NIR laser; Group 3: FeOxH–rGO; Group 4: FeOxH–rGO + NIR laser.

**Figure 6 nanomaterials-11-01947-f006:**
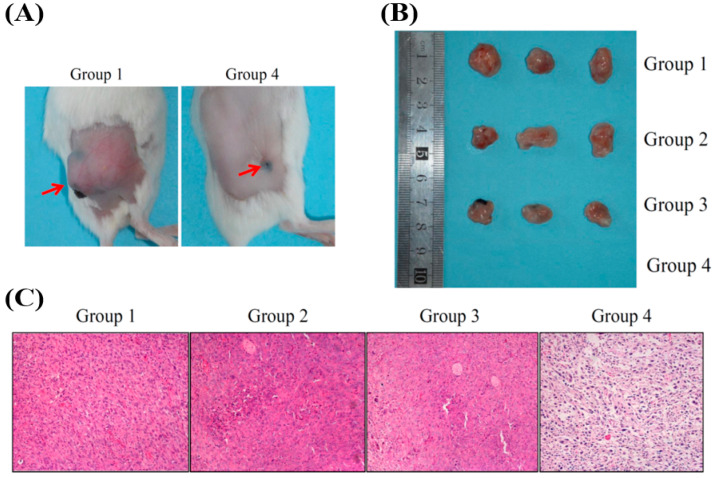
(**A**) Photographs of mice bearing 4T1 tumours 17 days after treatments. (**B**) Photographs showing the tumours from mice 17 days after intratumor injection of PBS and FeOxH–rGO, with or without NIR irradiation. (**C**) Haematoxylin and eosin staining of the tumour-tissue section after intratumor injection of PBS and FeOxH–rGO, with or without NIR irradiation. Group 1: PBS; Group 2: PBS + NIR laser; Group 3: FeOxH–rGO; Group 4: FeOxH–rGO + NIR laser. The condition of NIR laser irradiation was the same with Figure 5.

**Figure 7 nanomaterials-11-01947-f007:**
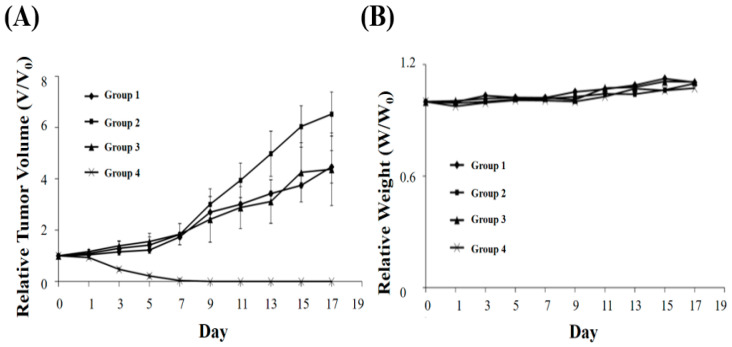
(**A**) In vivo tumour growth curves and (**B**) body weight curves in different groups of mice after various treatments indicated. The tumour volumes and body weight were normalised to their initial sizes and weight. Group 1: PBS; Group 2: PBS + NIR laser; Group 3: FeOxH–rGO; Group 4: FeOxH–rGO + NIR laser. Four groups of mice (n = 5 per group) were the same conditions with Figure 5 and Figure 6.

## Data Availability

The data that support the findings of this study are available from the corresponding author upon reasonable request.

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
