# Peer review of "Iron Hydroxide/Oxide-Reduced Graphene Oxide Nanocomposite for Dual-Modality Photodynamic and Photothermal Therapy In Vitro and In Vivo"

_nanomaterials, 2021, doi:10.3390/nano11081947_

Round 1
Reviewer 1 Report
Jane Chiu et al in their manuscript have synthesized Iron hydroxide/oxide, to absorb the near-infrared light and thus can be applied in Photothermal and phototherapeutic purposes.
The manuscript figures in general lack clarity and don’t add clear information that can be followed.
In Figure 1A, the TEM image doesn’t show the nanocomposite properties. The UV-Vis of the nanocomposite shows that the yield is very low and doesn’t give enough characteristics of the nanocomposite in the near-infrared region (Figure 1B). Figure 1C doesn’t add much information.
In Figure 2D. Why did the temperature drop down after 1000 seconds of irradiation ?? there is no explanation or discussion.
Figure 4A has shown that FeOxH–rGO nanocomposites without near-infra-red (NIR) are biocompatible and have no toxicity. However, their effects appear only when combined with NIR, but I don’t see any stability figure for the nanocomposite upon the NIR.
Figures 5 and 6 show that the tumors disappeared (burnt) in group 4 on day 2. However, there is no complete information about how did they calculate the nanocomposites doses that were injected intratumorally.
Author Response
We have revised the original manuscript and summarized the following changes (highlighted in the revised draft):
Reviewer 1
- In Figure 1A, the TEM image doesn’t show the nanocomposite properties.
Response: We apologize that we did not make this point very clearly in our original manuscript. We have added the Figure S1 in the supporting information. FeOxH–rGO nanocomposites was prepared by reaction of rGO (size ~ 300 nm; prepared from irradiation of GO with UV light for 5 h) (Figure S1) with Fe2+ in Tris–borate solution (5.0 mM, pH 7.0). When Fe2+ was prepared in Tris–borate solution (5.0 mM, pH 7.0), the iron(III) oxide-hydroxide (4Fe2+ + O2 + 6H2O / 4FeO(OH) + 8H+) and iron(II) hydroxide (Fe2+ + 2H2O / Fe(OH)2 + 2H+) were formed and immobilized on rGO. The FeOxH nanostructures were randomly distributed on the surface of rGO.
Figure S1. TEM images of rGO. rGO (size ~ 300 nm) was prepared from irradiation of GO with UV light for 5 h
- The UV-Vis of the nanocomposite shows that the yield is very low and doesn’t give enough characteristics of the nanocomposite in the near-infrared region (Figure 1B).
Response: We apologize that we did not make this point very clearly in our original manuscript. The absorbance of FeOxH-rGO was enhanced in the NIR window (Figure 1B), potentially as a result of FeOxH deepening the colour (Figure 1C) and increasing light absorption, suggesting FeOxH-rGO might possess high phototherapeutic potential in cancer treatment. We have also determined the photothermal conversion efficiency of FeOxH–rGO under NIR illumination. We found the estimated photothermal conversion efficiency of FeOxH–rGO to be 64.4%. GO-based material is capable of absorbing light and efficiently converting it to heat across a wide spectrum of wavelengths from UV to NIR, because of its delocalized electron arrangement. Laser illumination of GO-based material results in a rapid deoxygenation accompanied by an increase in thermal conductivity, and further enhance the heat-transfer process of GO-based material in the solution (1). Furthermore, rGO has been demonstrated as an effective photothermal conversion agent for photothermal therapy of tumor cells through NIR illumination (2). It seems that rGO plays a dual role in the photothermal process induced by FeOxH–rGO under NIR illumination, acting both as a thermal generator and as a thermal enhancer. We suggest the features of catalytic FeOxH may accelerate the deoxygenation of GO and improve the photothermal conversion efficiency of FeOxH–rGO nanocomposite (3).
- Figure 1C doesn’t add much information.
Response: The absorbance of FeOxH-rGO was enhanced in the NIR window (Figure 1B), potentially as a result of FeOxH deepening the colour (Figure 1C) and increasing light absorption, suggesting FeOxH-rGO might possess high phototherapeutic potential in cancer treatment (4).
- In Figure 2D. Why did the temperature drop down after 1000 seconds of irradiation ?? there is no explanation or discussion.
Response: The effects of the NIR laser on the photothermal conversion efficiency (the ratio of the absorption cross-section to the extinction cross-section) were studied by measuring the temperature increase of FeOxH-rGO compared with that of the medium. The experimental setup had a 10 mm-path-length quartz cuvette and an 808 nm laser that was used to irradiate the sample with a power density of 2.72 W/cm2 for 20 min (1200 seconds). Between experiments, the laser was turned off and allowed to cool to its initial temperature (temperature drop down) (Figure 2D). The photothermal conversion efficiency of FeOxH-rGO was estimated to be 64.4%.
- Figure 4A has shown that FeOxH–rGO nanocomposites without near-infra-red (NIR) are biocompatible and have no toxicity. However, their effects appear only when combined with NIR, but I don’t see any stability figure for the nanocomposite upon the NIR.
Response: We apologize that we did not make this point very clearly in our original manuscript. After NIR irradiation (5 min), at a concentration of 100 μM, Fe2+ ions immobilised on rGO showed great physiological stability (Figure S2B(g)). Furthermore, to test the stability of the photothermal heating, the temperature increase of the FeOxH–rGO under NIR irradiation was measured over two cycles of irradiation and cooling to room temperature. The thermal curves shown in Figure S3 indicate that the heating effect was extremely repeatable, demonstrating highly stable photothermal performance. More detailed information is added described in the Results and Discussion section (Line 160).
Figure S3. Photothermal stability of FeOxH–rGO nanocomposites (300 μg/mL) under NIR illumination.
- Figures 5 and 6 show that the tumors disappeared (burnt) in group 4 on day 2. However, there is no complete information about how did they calculate the nanocomposites doses that were injected intratumorally.
Response: We apologize that we did not make this point very clearly in our original manuscript. Each group of mice (n=5) was intratumorally injected with 20 μL PBS or FeOxH–rGO (2.4 mg/mL). More detailed information is added described in the Results and Discussion section (Line 229).

Reviewer 2 Report
This manuscript is interesting. I guess that this manuscript is worth to publish in this Journal. However, it is necessary to revise some points as follows.
- There is no description about particle size of Fe OxH-rGO. Authors should describe its particle size.
- Page 3, line 115: “Each group of mice were intratumorally injected with PBS or FeOxH-rGO” How volume authors injected into the tumor?
- In vivo and in vitro experiments, I understood efficiency of Fe OxH-rGO under the irradiation. However, there is no discussion about its mechanism. Is Fe OxH-rGO taken into the tumor cells? Or, Fe OxH-rGO is not taken into the tumor cells? It is important problem. Authors should discuss this point.
Author Response
Dear Editor,
Sub: Submission of revised manuscript
Manuscript ID: nanomaterials-1292474
Title: "Iron Hydroxide/Oxide-Reduced Graphene Oxide Nanocomposite for Dual-Modality Photodynamic and Photothermal Therapy in Vitro and in Vivo"
We thank you for considering our research work for peer review and publication. The response to each comment is presented below this letter. Kindly consider our revised manuscript for publication in Nanomaterials.
.
.
Sincerely,
Chia-Hua Lin et al.
We have revised the original manuscript and summarized the following changes (highlighted in the revised draft):
- There is no description about particle size of Fe OxH-rGO. Authors should describe its particle size.
Response: We apologize that we did not make this point very clearly in our original manuscript. We have added the Figure S1 in the supporting information. FeOxH–rGO nanocomposites was prepared by reaction of rGO (size ~ 300 nm; prepared from irradiation of GO with UV light for 5 h) (Figure S1) with Fe2+ in Tris–borate solution (5.0 mM, pH 7.0). When Fe2+ was prepared in Tris–borate solution (5.0 mM, pH 7.0), the iron(III) oxide-hydroxide (4Fe2+ + O2 + 6H2O / 4FeO(OH) + 8H+) and iron(II) hydroxide (Fe2+ + 2H2O / Fe(OH)2 + 2H+) were formed and immobilized on rGO. The FeOxH nanostructures were randomly distributed on the surface of rGO.
- Page 3, line 115: “Each group of mice were intratumorally injected with PBS or FeOxH-rGO” How volume authors injected into the tumor?
Response: We apologize that we did not make this point very clearly in our original manuscript. Each group of mice (n=5) was intratumorally injected with 20 μL PBS or FeOxH–rGO (2.4 mg/mL). More detailed information is added described in the Results and Discussion section (Line 229).
- In vivo and in vitro experiments, I understood efficiency of Fe OxH-rGO under the irradiation. However, there is no discussion about its mechanism. Is Fe OxH-rGO taken into the tumor cells? Or, Fe OxH-rGO is not taken into the tumor cells? It is important problem. Authors should discuss this point.
Response: We appreciate the constructive suggestion. More detailed information is added described in the Results and Discussion section (Line 217-218).
Reference:
- Abdelsayed, V.; Moussa, S.; Hassan, H. M.; Aluri, H. S.; Collinson, M. M.; El-Shall, M. S. Photothermal Deoxygenation of Graphite Oxide with Laser Excitation in Solution and Graphene-Aided Increase in Water Temperature. Phys. Chem. Lett. 2010, 1, 2804‒2809.
- Akhavan, O.; Ghaderi, E.; Aghayee, S.; Fereydooni, Y.; Talebi, A. The use of a glucose-reduced grapheme oxide suspension for photothermal cancer therapy. Mater. Chem. 2012, 22, 13773‒13781.
- Zedan, A. F.; Moussa, S.; Terner, J.; Atkinson, G.; El-Shall, M. S. Ultrasmall Gold Nanoparticles Anchored to Graphene and Enhanced Photothermal Effects by Laser Irradiation of Gold Nanostructures in Graphene Oxide Solutions. ACS Nano 2013, 7, 627‒636.
- Yu, H.H.; Lin, C.H.; Chen, Y.C.; Chen, H.H.; Lin, Y.J.; Lin, K.A. Dopamine-modified zero-valent iron nanoparticles for dual-modality photothermal and photodynamic breast cancer therapy. ChemMedChem 2020, 15, 1645-1651.